# Fractional Calculus-Based Statistical Damage Model of Unsaturated Soil under the Coupling Effect of Moistening and Stress Fields

**Hua Zhang** [1,2] and **Peng Wang** [3,4,*]

1 School of Architecture and Civil Engineering, Chengdu University, Chengdu 610106, China; chalkz@126.com
2 Sichuan Provincial Engineering Research Center of City Solid Waste Energy and Building Materials Conversion and Utilization Technology, Chengdu University, Chengdu 610106, China
3 Geotechnical Engineering Institute, Sichuan Institute of Building Research, Chengdu 610081, China
4 Sichuan Engineering Research Center for Mechanical Properties and Engineering Technology of Unsaturated Soils, Chengdu University, Chengdu 610106, China
* Correspondence: wrypscre@163.com

**Abstract:** Unsaturated soil exhibits extremely complex engineering mechanical properties under the coupling effect of moistening and stress fields. Firstly, the effective stress principle and limit equilibrium conditions of unsaturated soil under the coupling effect of moistening and stress fields were discussed based on the basic principles of unsaturated soil. Secondly, a fractional-order model considering the viscoelasticity and strain hardening of unsaturated soil was established based on the fractional calculus theory. Then, based on the principle of damage mechanics, the damage variable evolution equation under the coupling effect of moistening and stress fields was established, and the fractional calculus-based statistical constitutive damage model of unsaturated soil under the coupling effect of moistening and stress fields was developed. In turn, parameters of the developed model were solved using a triaxial test of unsaturated loess, and the calculated data using the developed model were compared with the experimental data, which demonstrated that the developed model in this paper performed well in describing the whole strain hardening process of unsaturated soil under the coupling effect of moistening and stress fields. Finally, the sensitivity of the main parameters of the developed model was discussed under the coupling effect of moistening and stress fields, which showed that the proposed model performed well in reflecting the main mechanical properties of unsaturated loess.

**Keywords:** unsaturated soil; moistening field; stress field; viscoelasticity; strain hardening; damage model

## 1. Introduction

With the progress of urbanization in China, an increasing number of construction projects of various types are implemented in regions with unsaturated soil, such as side slope projects, subgrade projects, and retaining wall projects [1,2], where the control of strength and deformation has been a key issue. Water and load are two important factors affecting the strength and deformation of unsaturated soil. The distribution of water in unsaturated soil constitutes a moistening field on the semi-infinite plan, where hydraulic effects are generated [3,4] that affect the strength and deformation of the soil. The load transferring through the soil particles forms a stress field that directly affects the strength and deformation of the soil, which is, in essence, the effective stress principle of unsaturated soil [5,6]. Therefore, clarifying the hydraulic properties and effective stress principles of unsaturated soil is an important prerequisite for analyzing soil strength and deformation.

Up to now, the hydraulic properties of unsaturated soil are mainly studied based on soil-water property tests at the macroscopic level. Li et al. [7] plotted the soil-water characteristic curve (SWCC) of two types of loess using pressure plate and filter paper tests,

then proposed a method to predict SWCC by mercury intrusion porosimetry (MIP) with the contact angle (CA) as a fitting coefficient. Satyanaga et al. [8] adopted the evaporation method and the chilled-mirror dew-point method to measure SWCC in the large suction range and used shrinkage curves to correct for soil deformation. Alves et al. [9] developed a theoretical model for predicting SWCC using pore scale analysis, three-dimensional approximation of pore geometry, and the concept of unit cells. Sillers et al. [10] analyzed some mathematical models of SWCC from various perspectives, such as the number of parameters and whether the parameters have physical significance, and found that three-parameter models performed well in reflecting SWCC. Relevant studies have also analyzed the factors of SWCC curves. Jiang et al. [11] concluded that the initial dry density affected the pore size and number, which in turn affected the SWCC of loess. The strength of unsaturated soil has been a hot issue in unsaturated soil research. Many studies [12–16] considered the adsorption strength in the shear strength theory of saturated soil as the shear strength of unsaturated soil, which inherited the advantage of saturated soil mechanics and introduced the effective stress principle into the unsaturated soil theory [17]. Some studies [18,19] have also predicted the shear strength of unsaturated soil by SWCC, which indicates an inherent coupling correlation between the moistening field and stress field in the unsaturated soil structure.

Soil is a typical viscoelastic material [20–22] that has a certain sensitivity to the genesis of soils, stress history, stress level, stress path, and water content [23]. Soil composition and soil non-saturation can be reflected by the soil–water characteristic curve, that is, different soil composition shows different matric suction under the same water content [11,24]. Matrix suction directly affects the stress–strain relationship of unsaturated soil [11]. Strain hardening is a typical stress–strain relationship for unsaturated soil and an important reflection of its mechanical properties. Shao et al. [25] analyzed the structural properties of unsaturated loess based on triaxial tests and developed a constitutive model reflecting the strain hardening of unsaturated loess based on water content and stress state. Li et al. [24] studied the volumetric variation and hydraulic properties of unsaturated soil at different water contents and found a logarithmic relationship between the hardening coefficient and saturation of unsaturated soil. In the meantime, the soil is a typical structural material, and strain hardening directly affects its strength and deformation. Therefore, constitutive models of unsaturated soil have been developed from the perspective of damage. Shi et al. [26] concluded that the hardening and damage of soil were attributed to the closure and expansion of pores and established a damage model for frozen soil. Xu et al. [27] developed a soil damage constitutive model considering suction effects, plastic flow, and damage based on the continuous damage theory. Yao et al. [28] adopted the novel electrical impulse method and water-air transport laws to develop a percolation model for collapsibility elastoplasticity damage that considered the loess structure based on the Barcelona model. Moreover, it has been found that fractional calculus is a powerful tool for modeling the viscoelastic/viscoplastic behaviors and is particularly suited for building a time-dependent constitutive model [29–33]. Based on fractional calculus, Yin et al. [31] presented the fractional Bingham model that can describe the time dependent behavior in muddy clay with yield strength. By replacing a Newtonian dashpot in the classical Nishihara model with the fractional derivative Abel dashpot, Zhou et al. [32] proposed a new creep constitutive model for salt rock on the basis of a time-based fractional derivative. Using experimental data from several tests, such as creep and creep recovery, that were performed at different temperatures and at different stress levels, Di Mino et al. [33] presented a fractional viscoelastic and viscoplastic model of asphalt mixtures. With various angles, reasonable methods, clear patterns, and reliable conclusions, research on the mechanical properties of unsaturated soil provided theoretical and technical support to the relevant engineering projects and strongly promoted the development of unsaturated soil mechanics.

However, deficiencies in the study of mechanical properties of unsaturated soil are non-negligible [34]. For example, relatively few studies have focused on the laws of mechanical properties, viscoelasticity, and damage evolution of unsaturated soil under the

coupling effect of hydraulic properties and stress, and no consensus has been reached. In the meantime, these properties are the key to solving the strength and deformation problems in regions with unsaturated soil. Therefore, it is necessary to further study the mechanical properties of unsaturated soil. Based on the basic theories of soil mechanics, fractional-order calculus and statistical damage principle, a fractional-order damage constitutive model of coupled moistening fields and stress fields is established by determining the effective stress, shear strength expression, viscoelastic expression, and damage variable evolution formula of unsaturated soil. The hydraulic effect, strength law, viscoelastic law, and damage evolution law of unsaturated soil in moistening fields and stress fields are further analyzed through the discussion of model parameters. This study provides a new model to describe the mechanical mechanism of strain hardening of unsaturated soil under the influence of water and stress.

## 2. Principle and Intensity of Effective Stress of Unsaturated Soil

### 2.1. Principle of Effective Stress in Unsaturated Soil

In its natural state, unsaturated soil is under the effect of moistening and stress fields. Typical unsaturated soil consists of three phase media: soil particles and water and air in the pores. The phase media share the load and interact with each other. The principle of effective stress in unsaturated soil is shown in Figure 1. On any section $A$-$A$ of the soil with cross-sectional area $A$, a total load $\sigma$ is applied, and the water content is increased by $\Delta A_{\mathrm{sw}}$. Assuming the total area of soil particles is $A_{\mathrm{s}}$, $A_{sw0}$ and $A_{sw1}$ represent the cross-sectional area of soil particles in pore water before and after the change in the soil water content, respectively, and $A_{sa0}$ and $A_{sa1}$ denote the cross-sectional area of soil particles in the pore air before and after the change of soil water content, respectively. In addition, $A_{\mathrm{s}} = A_{\mathrm{sw0}} + A_{\mathrm{sa0}} = A_{\mathrm{sw1}} + A_{\mathrm{sa1}}$. Assuming the total pore area is $A_{v}$, $A_{w0}$ and $A_{w1}$ denote the cross-sectional area of pore water before and after the change in the soil water content, respectively, and $A_{a0}$ and $A_{a1}$ denote the cross-sectional area of pore air before and after the change of soil water content, respectively. In addition, $A_{\mathrm{v}} = A_{\mathrm{w0}} + A_{\mathrm{a0}} = A_{\mathrm{w1}} + A_{\mathrm{a1}}$. Then, $A = A_{\mathrm{s}} + A_{\mathrm{v}}$. The soil skeleton stresses before and after the change of soil water content are $\sigma_{sv0}$ and $\sigma_{sv1}$, the pore water pressure is $u_{w0}$ and $u_{w1}$, and the pore air pressure is $u_{a0}$ and $u_{a1}$.

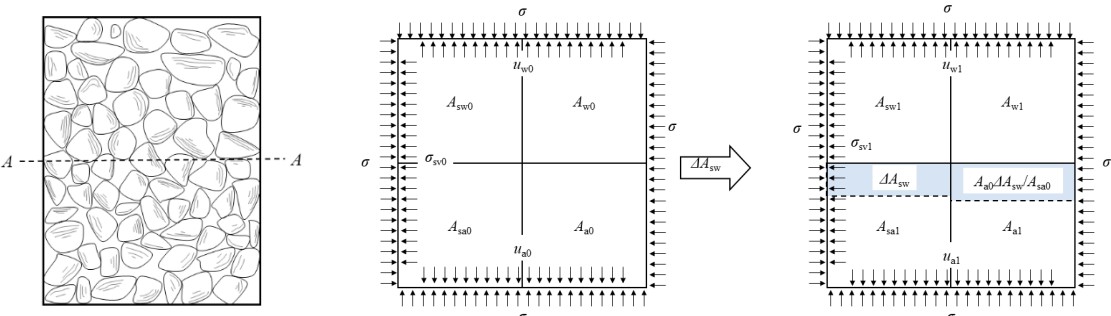

**Figure 1.** Principle of effective stress in unsaturated soil.

The relationship between the soil particle cross-sectional area before and after the change in water content is:

$$\begin{cases} A_{\mathrm{sw1}} = A_{\mathrm{sw0}} + \Delta A_{\mathrm{sw}} \\ A_{\mathrm{sa1}} = A_{\mathrm{sa0}} - \Delta A_{\mathrm{sw}} \end{cases} \tag{1}$$

The pore water cross-sectional area and pore air cross-sectional area before and after the change in water content have the following relationships:

$$\begin{cases} A_{w1} = A_v - A_{a0}\left(1 - \frac{\Delta A_{sw}}{A_{sa0}}\right) \\ A_{a1} = A_{a0}\left(1 - \frac{\Delta A_{sw}}{A_{sa0}}\right) \end{cases} \tag{2}$$

Assuming that the pore water and pore air are connected, and that the three-phase media are homogeneous in the soil, then:

$$\begin{cases} \frac{A_{sw0}}{A_s} = \frac{A_{w0}}{A_v} = S_{r0} \\ \frac{A_{sw1}}{A_s} = \frac{A_{w1}}{A_v} = S_{r1} \end{cases} \tag{3}$$

where $S_{r0}$ and $S_{r1}$ denote the saturation of the soil before and after the change in the water content, respectively, and $S_{r1} = S_{r0} + \Delta S_r$.

According to Figure 1 and the above analysis, the equilibrium equation after the change of water content in the *A-A* section is:

$$\sigma A = \sigma'_{sv}A_{s1} + u_{w1}(A_{sw1} + A_{w1}) + u_{a1}(A_{sa1} + A_{a1}) \tag{4}$$

It can be deduced from Equations (1)–(4) that:

$$\sigma = \sigma'_{sv}\frac{A_{s1}}{A} + u_{w1}S_{r1} + u_{a1}(1 - S_{r1}) \tag{5}$$

where $n$ denotes the porosity of the soil. $\Delta S_r$ denotes the incremental saturation.

Assuming that $\sigma' = \sigma'_{sv}\frac{A_{s1}}{A} = \sum P'_{svi}/A$, where $\sigma'$ denotes the effective stress, it can be brought into Equation (5) to give the following after sorting and transformation.

$$\sigma' = (\sigma - u_{a1}) + (u_{a1} - u_{w1})S_{r1} \tag{6}$$

According to Equation (6), the effective stress of unsaturated soil consists of the net confining pressure $(\sigma - u_{a1})$ and the matrix suction $(u_{a1} - u_{w1})S_{r1}$.

The following can be obtained from VG model transformation:

$$s = \delta\left[(S_{r0} + \Delta S_r)^{\beta/(1-\beta)} - 1\right]^{1/\beta} \tag{7}$$

where $s$ denotes the matrix suction and $s = u_a - u_w$. $\delta$ and $\beta$ can be obtained through fitting by the soil–water properties test.

The effective stress principle of unsaturated soil expressed by saturation can be obtained by substituting Equation (7) into (6).

$$\sigma' = (\sigma - u_a) + \delta S_{r1}\left[S_{r1}^{\beta/(1-\beta)} - 1\right]^{1/\beta} \tag{8}$$

### 2.2. The Shear Strength Theory of Unsaturated Soil

Based on the effective stress principle of unsaturated soil, the shear strength in the triaxial test under the moistening and stress fields can be obtained through Schrefler's [15] Formula (16) for shear strength:

$$\tau = c' + (\sigma - u_a)\tan\varphi' + \delta S_{r1}^2\left[S_{r1}^{\beta/(1-\beta)} - 1\right]^{1/\beta}\tan\varphi' \tag{9}$$

where $c'$ and $\varphi'$ are the effective cohesion and effective internal friction angle of saturated soil, respectively.

From Equation (9), the limit equilibrium condition for unsaturated soil under triaxial test stress conditions can be obtained as follows:

$$\sigma_1 = \sigma_3 \tan^2(45 + \frac{\varphi'}{2}) + 2c_{\text{tatal}} \tan(45 + \frac{\varphi'}{2}) \tag{10}$$

where $c_{\text{total}} = c' + \delta S_{r1}^2 [S_{r1}^{\beta/(1-\beta)} - 1]^{1/\beta} \tan \varphi'$.

### 2.3. Strain Characteristics of Unsaturated Soil in the Stress Field

A large number of tests have shown that the strain hardening curves for different types of unsaturated soil have different trends under the same confining pressure. As shown in Figure 2, the deviatoric stresses of the three curves are the same at the time of destruction, which means that the three curves have the same Mohr's stress circle. Therefore, the hardening capacity of the three strain hardening curves must be reflected in order to distinguish the engineering properties of the soil.

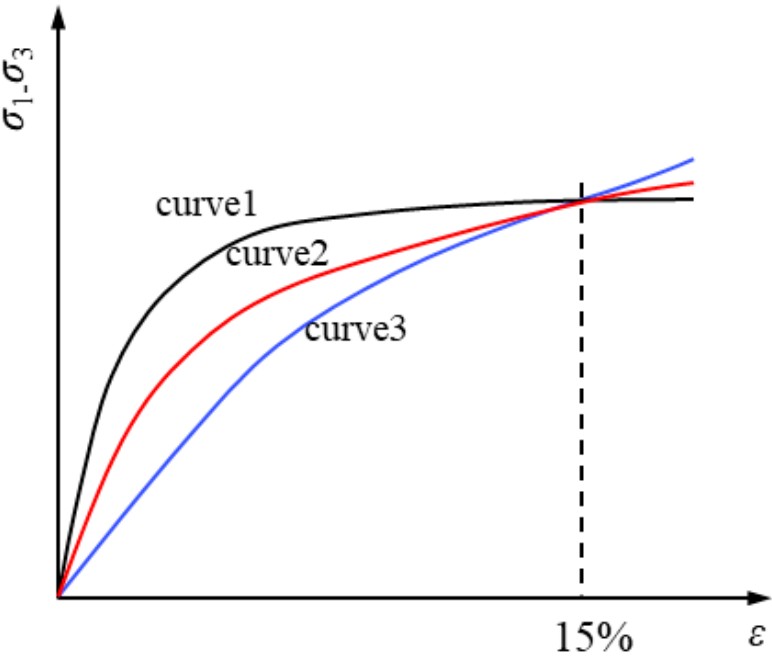

**Figure 2.** Stress–strain curves of different unsaturated soil under the same confining pressure.

Since soil is a typical viscoelastic material, the fractional calculus theory is introduced to effectively describe the strain hardening and viscoelasticity of unsaturated soil. The differential operators from fractional calculus and the stress history of memory materials are widely used to describe the mechanical properties of viscoelastic materials. According to the Riemann–Liouville fractional calculus operator theory, the integral of function $f(t)$ of order $\alpha$ is defined as:

$$\frac{\mathrm{d}^{-\alpha} f(t)}{\mathrm{d}t^{-\alpha}} =_{t0} D_t^{-\alpha} f(t) = \frac{1}{\Gamma(\alpha)} \int_a^t (t-x)^{\alpha-1} f(x)dx \tag{11}$$

The fractional order differentiation is defined as:

$$_{t0}D_t^{\alpha} f(t) = \frac{\mathrm{d}^{\alpha} f(t)}{\mathrm{d}t^{\alpha}} = \frac{\mathrm{d}^n}{\mathrm{d}t^n} \left[ D^{-(n-\alpha)} f(t) \right] \tag{12}$$

where $0 < \alpha$ is the fractional order. $\Gamma(*)$ is a Gamma function defined as: $\mathrm{Re}(z) > 0$, $\Gamma(\tau) = \int_0^\infty e^{-x} t^{\tau-1} dx$.

Therefore, the strain hardening relationship of unsaturated soil can be derived from the fractional calculus theory as the following:

$$\sigma(t) = E_0 D^\alpha \varepsilon(t) \tag{13}$$

where $E_0$ denotes the initial elastic modulus, $0 < \alpha < 1$ and a larger $\alpha$ means a smaller strain hardening capacity of the soil,

Generally, constant strain rate loading is adopted in unsaturated soil shear tests, and the strain–time relationship is:

$$t = \frac{\varepsilon}{v_0} \tag{14}$$

where the constant $v_0$ denotes the strain rate.

Bringing Equation (14) into Equation (13), the strain hardening viscoelasticity model in the triaxial test is obtained as follows:

$$\sigma = E_0 v_0^\alpha \frac{\varepsilon^{(1-\alpha)}}{\Gamma(2-\alpha)} \tag{15}$$

*2.4. The Damage Evolution Law of Unsaturated Soil*

2.4.1. Micro-Unit Damage Variables

Under a certain load, the micro-units of unsaturated soil are damaged when the stress level reaches the strength limit. Assuming that the total number of soil micro-units is $S_t$ and the number of damaged micro-units is $S_f$, the damage variables can be defined as:

$$D_{\mathrm{d}} = \frac{S_{\mathrm{f}}}{S_{\mathrm{t}}} \tag{16}$$

Assuming that $S_f$ is large enough and the micro-unit damage law is in Laplace distribution, then the probability density function is:

$$\varphi(F|m_0, K_0) = \frac{1}{2K_0} \exp\left(-\frac{F_{\mathrm{d}} - m_0}{K_0}\right) \tag{17}$$

where $F_d$ is a randomly distributed variable representing the strength of soil micro-units and $m_0$ and $K_0$ are Laplace distribution parameters.

Assuming that the range of micro-unit strength is $[F_{\mathrm{d}}, F_{\mathrm{d}} + \mathrm{d}F_{\mathrm{d}}]$ and the micro-units are damaged, it can be obtained from Equations (6) and (17) that:

$$D_{\mathrm{d}} = \int_{-\infty}^{F_{\mathrm{d}}} P(x)\mathrm{d}x = 1 - \frac{1}{2}\exp\left(-\frac{F_{\mathrm{d}} - m_0}{K_0}\right) \tag{18}$$

2.4.2. Evolution of Damage Variables

Water content and load are two important external factors affecting the mechanical properties of unsaturated soil. Therefore, the evolution of damage variables can be divided into two parts: moistening field damage caused by changes in soil water content and stress field damage caused by soil load levels.

Under natural conditions, unsaturated soil has initial moistening and stress fields, and it is assumed that damage evolution of unsaturated soil structure or mechanical properties occurred first in the moistening field and then in the stress field. The damage evolution of an isolated soil sample is shown in Figure 3. Among them, $w_0$, $w$, and $\tilde{w}$ are the water contents of the soil samples in the normal state, after moistening field damage, and in the equivalent undamaged state, respectively. $\sigma_0$, $\sigma_{\mathrm{w}}$, $\sigma$, and $\tilde{\sigma}$ are the effective stresses of the soil samples in the normal state, after moistening field damage, after stress field damage, and in the equivalent undamaged state, respectively. $E_0$, $E_{\mathrm{w}}$, and $E_\sigma$ are the elastic modulus of the soil samples in the normal state, after moistening field damage, and after

stress field damage, respectively. $dA_0$, $dA_w$, $dA_\sigma$, and $d\tilde{A}$ are the area of the soil samples in the normal state, after moistening field damage, after stress field damage, and in the equivalent undamaged state, respectively.

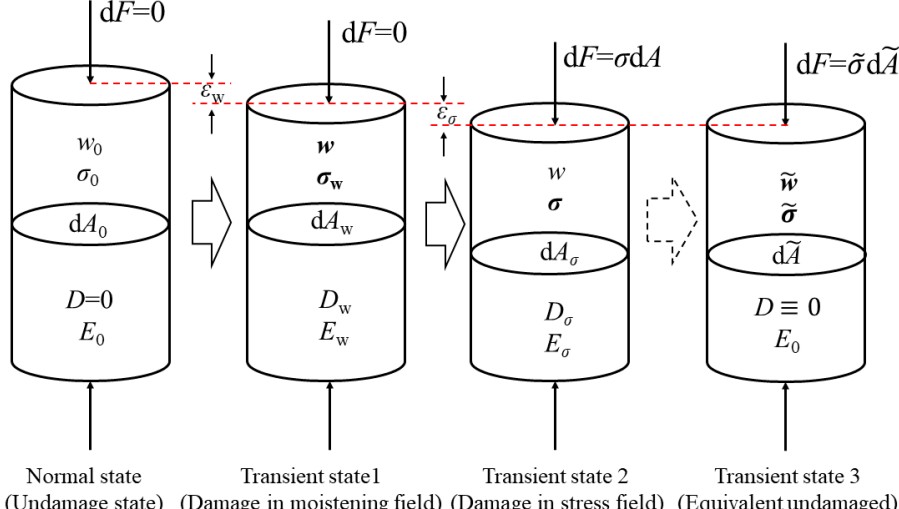

**Figure 3.** Damage evolution mechanism of unsaturated soil under the coupling effect of moistening and stress fields.

(a) As the soil sample transitioned from the normal state to the transient state 1, the moistening field changed, and the water content changed from $w_0$ to $w$. The following relationships are found within the soil sample:

$$\sigma_0 A_0 = \sigma_w A_w \tag{19}$$

$$D_w = \frac{A_0 - A_w}{A_0} \tag{20}$$

It can be deduced from Equations (19), (20) and (15) that:

$$E_w = E_0(1 - D_w) \tag{21}$$

$$\sigma_w = E_0(1 - D_w)v_0^\alpha \frac{\varepsilon_w^{(1-\alpha)}}{\Gamma(2-\alpha)} \tag{22}$$

The above equation is the damage constitutive model after the changes of water content in the moistening field.

(b) As the soil sample transitioned from transient state 1 to transient state 2, the stress field changed, and the stress changed from $\sigma_w$ to $\sigma$. The following relationships are found within the soil sample:

$$\sigma_w A_w = \sigma A_\sigma \tag{23}$$

$$D_w = \frac{A_w - A_\sigma}{A_w} \tag{24}$$

It can be deduced from Equations (23), (24) and (15) that:

$$E_\sigma = E_w(1 - D_\sigma) \tag{25}$$

$$\sigma = E_{\text{w}}(1 - D_\sigma)v_0^\alpha \frac{\varepsilon_\sigma^{(1-\alpha)}}{\Gamma(2-\alpha)} \tag{26}$$

The above equation is the damage constitutive model after the changes of stress in the stress field.

The damage variables under the coupling effect of moistening and stress fields can be obtained from Equations (21), (25) and (26):

$$\sigma = E_0(1 - D_{\text{w},\sigma})v_0^\alpha \frac{\varepsilon_\sigma^{(1-\alpha)}}{\Gamma(2-\alpha)} \tag{27}$$

$$D_{\text{w},\sigma} = D_{\text{w}} + D_\sigma - D_{\text{w}}D_\sigma \tag{28}$$

The statistical damage model can be deduced from Equations (18) and (27):

$$\sigma = \frac{1}{2}E_0 \cdot \exp\left(-\frac{F_{\text{d}} - m_0}{K_0}\right) \cdot v_0^\alpha \frac{\varepsilon^{(1-\alpha)}}{\Gamma(2-\alpha)} \tag{29}$$

Equation (29) is the fractional calculus-based statistical damage constitutive model under uniaxial shear conditions.

Under triaxial test stress conditions:

$$\sigma_i = \sigma'_i(1 - D_{\text{w},\sigma}) \tag{30}$$

$$\sigma'_i = E_0(1 - D_{\text{w},\sigma})v_0^\alpha \frac{\varepsilon_i^{(1-\alpha)}}{\Gamma(2-\alpha)} + \mu(\sigma'_j + \sigma'_k) \tag{31}$$

It can be deduced from Equations (18), (30) and (31) that:

$$\sigma_1 = \frac{1}{2}E_0 \cdot \exp\left(-\frac{F_{\text{d}} - m_0}{K_0}\right)v_0^\alpha \frac{\varepsilon_1^{(1-\alpha)}}{\Gamma(2-\alpha)} + 2\mu\sigma_3 \tag{32}$$

Equation (32) is the fractional calculus-based statistical damage constitutive model under triaxial shear conditions.

*2.5. The Micro-Unit Strength of Unsaturated Soil*

According to Equation (32), determining the micro-unit strength $F_{\text{d}}$ is the key to solving the model. Under triaxial shearing test stress conditions, the micro-unit strength is:

$$F_{\text{d}} - f(\widetilde{\sigma}) = 0 \tag{33}$$

where $\widetilde{\sigma}$ is the effective stress of the micro-unit. In addition, the micro-unit is deconstructed when $F_{\text{d}} - f(\widetilde{\sigma}) \leq 0$.

It can be deduced from Equations (9), (10) and (33) that:

$$F_{\text{d}} = \left(\widetilde{\sigma}_1 - u_a\right)(1 - \sin\varphi) - 2c_{\text{total}} \cdot \cos\varphi - \left(\widetilde{\sigma}_3 - u_a\right)(1 + \sin\varphi) \tag{34}$$

According to the equivalent undamaged relationship for transient state 2:

$$\widetilde{\sigma} = \frac{\sigma}{1 - D_{\text{w},\sigma}} \tag{35}$$

From Equation (35), the effective stress under triaxial shearing test stress conditions can be obtained as follows:

$$\widetilde{\sigma}_i = \frac{\sigma_i}{1 - D_{\text{w},\sigma}} \tag{36}$$

where $i$ = 1, 2, 3.

Given Equations (31), (36), and $\sigma_2 = \sigma_3$, under triaxial shearing test stress conditions:

$$D_{\mathrm{w},\sigma} = 1 - \frac{\Gamma(2-\alpha)}{E_0 v_0^\alpha} \frac{(\sigma_1 - 2\mu\sigma_3)}{\varepsilon_i^{(1-\alpha)}} \tag{37}$$

Thus, it can be deduced from Equations (35), (37) and (38) that:

$$F_{\mathrm{d}} = \frac{E_0 v_0^\alpha \varepsilon_1^{(1-\alpha)}}{(\sigma_1 - 2\mu\sigma_3)\Gamma(2-\alpha)} \times \left[(\sigma_1 - \sigma_3) - (\sigma_1 + \sigma_3)\sin\varphi\right] + 2u_a \sin\varphi - 2c_{\mathrm{total}}\cos\varphi \tag{38}$$

## 3. Model Parameter Solving

The proposed damage model has 10 parameters, where $\mu$ is determined by basic geotechnical tests, $\delta$ and $\beta$ can be obtained through fitting by soil-water properties tests, $c'$ and $\varphi'$ can be determined by conventional triaxial tests on saturated soil, and $E_0$, $v_0$, $\alpha$, $m_0$, and $K_0$ can be determined by triaxial tests on unsaturated soil. The specific solutions of $m_0$ and $k_0$ are as follows:

It can be deduced by combining Equations (18) and (37) that:

$$F_{\mathrm{d}} = m_0 - K_0 \ln(M) \tag{39}$$

where $M = \frac{(2\sigma_1 - 4\mu\sigma_3)\Gamma(2-\alpha)}{E_0 v_0^\alpha \varepsilon_1^{(1-\alpha)}}$.

## 4. Model Validation and Discussion

### 4.1. Model Parameters and Validation

Triaxial series tests for unsaturated soils on Xi'an loess, performed by Gao et al. [35], were selected to verify the ability of the proposed model to describe the mechanical behavior of unsaturated soil loess. The details of the soils and tests can be found in Gao et al. [35]. The values of model parameters identified for Xi'an loess are given in Table 1. The test apparatus used in this study was the unsaturated soil triaxial test equipment developed by Professor Chen Zhenghan, which was calibrated before the test. The initial matrix suction of the samples was adjusted to 50 kPa, 100 kPa, and 200 kPa, and the confining pressure was adjusted to 100 kPa, 200 kPa, and 300 kPa, respectively. The consolidation stability standard of unsaturated soil is 0.005 mL/h, and the average consolidation time of each sample is greater than 40 h. The shear rate was set to 0.0072 mm/min, and it took about 30 h to reach an axial strain of 15%. The results of the triaxial shearing test for unsaturated soil are shown in Figure 4.

**Table 1.** Basic physical properties of Xi'an loess collected from Yan'an New District (Data from Gao et al. [35]).

| Dry Density $\rho_{\mathrm{d}}$/g·cm$^{-3}$ | Relative Density $d_{\mathrm{s}}$ | Plastic Limit $w_{\mathrm{p}}$/% | Liquid Limit $w_{\mathrm{L}}$/% |
|---|---|---|---|
| 1.51 | 2.71 | 17.3 | 31.1 |

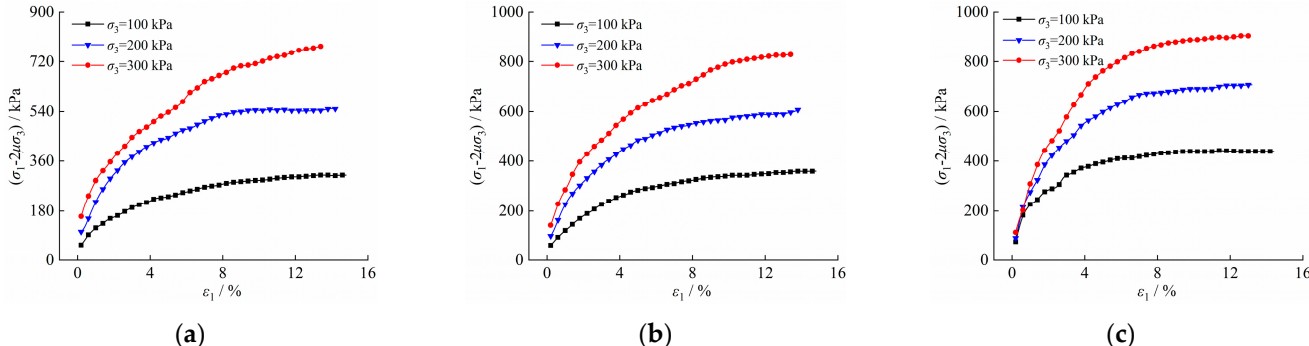

**Figure 4.** Triaxial test curve of unsaturated loess from Yan′an New District (Gao et al. [35]). (**a**) $s$ = 50 kPa; (**b**) $s$ = 100 kPa; (**c**) $s$ = 200 kPa.

According to the triaxial shearing test results of unsaturated loess from Yan′an New District, shown in Figure 4, the stress–strain relationships are all strain-hardening, which increases with the increase of matrix suction and confining pressure. The increased shear strength and slope of the stress–strain curve of unsaturated loess indicates enhanced viscoelasticity and strain-hardening of unsaturated loess.

As shown in Table 2, each parameter in the proposed model is solved according to the test results of unsaturated loess, and parameters $m_0$ and $K_0$ are solved as shown in Figure 5.

**Table 2.** Parameters of the proposed model.

| $s$ /kPa | $\sigma_3$ /kPa | Basic Physical Parameters | | | Saturated Soil Strength Parameters | | Laplace Distribution Parameters | | Strain Hardening Parameters |
|---|---|---|---|---|---|---|---|---|---|
| | | $E_0$ /kPa | $\mu$ | $v_0$ /mm·h$^{-1}$ | $c'$ /kPa | $\phi'$ /° | $m_0$ | $K_0$ | $\alpha$ |
| 50 | 100 | 19.7 | | | | | 122.03 | 273.50 | 0.452 |
| | 200 | 20.5 | | | | | 168.20 | 227.05 | 0.330 |
| | 300 | 20.9 | | | | | 246.61 | 361.40 | 0.270 |
| 100 | 100 | 25.7 | 0.38 | 0.432 | 12.26 | 30.65 | 128.11 | 310.78 | 0.435 |
| | 200 | 27.1 | | | | | 174.51 | 317.17 | 0.325 |
| | 300 | 28.0 | | | | | 220.66 | 470.55 | 0.251 |
| 200 | 100 | 31.0 | | | | | 227.36 | 396.41 | 0.255 |
| | 200 | 31.8 | | | | | 283.92 | 441.97 | 0.219 |
| | 300 | 32.6 | | | | | 332.64 | 493.81 | 0.170 |

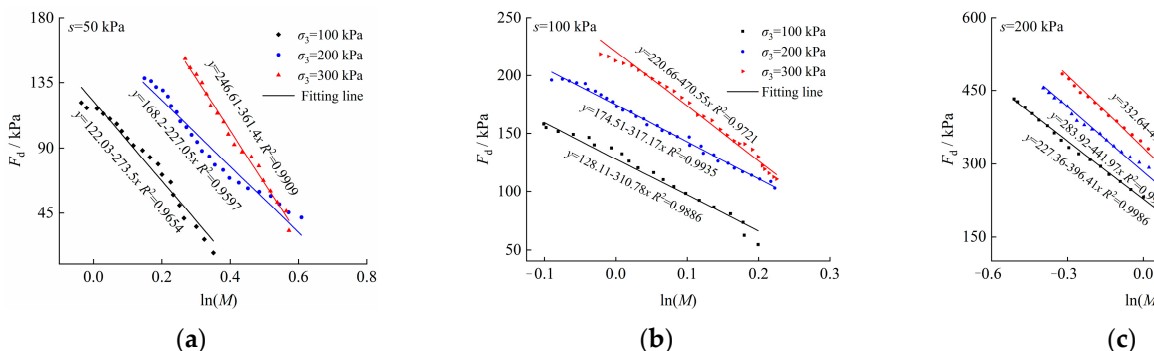

**Figure 5.** Fitting lines for parameters $m_0$ and $K_0$. (**a**) $s$ = 50 kPa; (**b**) s = 100 kPa; (**c**) $s$ = 200 kPa.

As shown in Figure 6, based on the calculated parameters, the results of the proposed model were compared with the experimental data for validation.

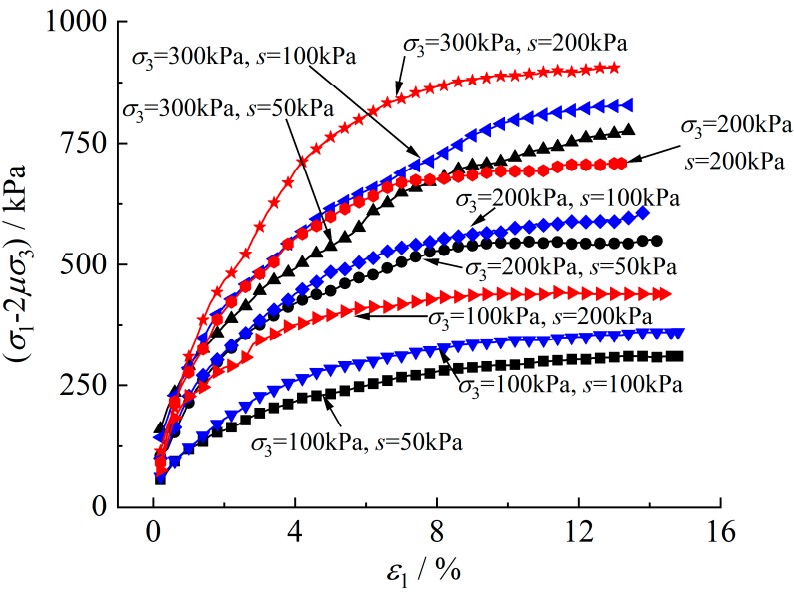

**Figure 6.** Comparison between the proposed model and experimental data.

According to Figure 6, the slope and intensity of the initial viscoelastic phase increased with the increase in matrix suction, indicating significant effects of the moistening field on the mechanical properties of unsaturated soil. The hardening capacity and strength of unsaturated soil increased with the increase in confining pressure, indicating that the change in the stress field has a significant effect on the mechanical properties of unsaturated soil. Therefore, the model proposed in this study performed well in describing the triaxial shearing strain-hardening test data of unsaturated soil.

For the proposed model, mechanical properties of unsaturated soil under 100~300 kPa confining pressure are verified in this paper. For the mechanical properties of unsaturated soil under other confining pressures, more test data are needed for further verification.

*4.2. Discussion*

(1)　Damage variable *D*

The variation pattern of damage variable *D* under the effect of matrix suction and confining pressure obtained from experiments and model analysis is shown in Figure 7.

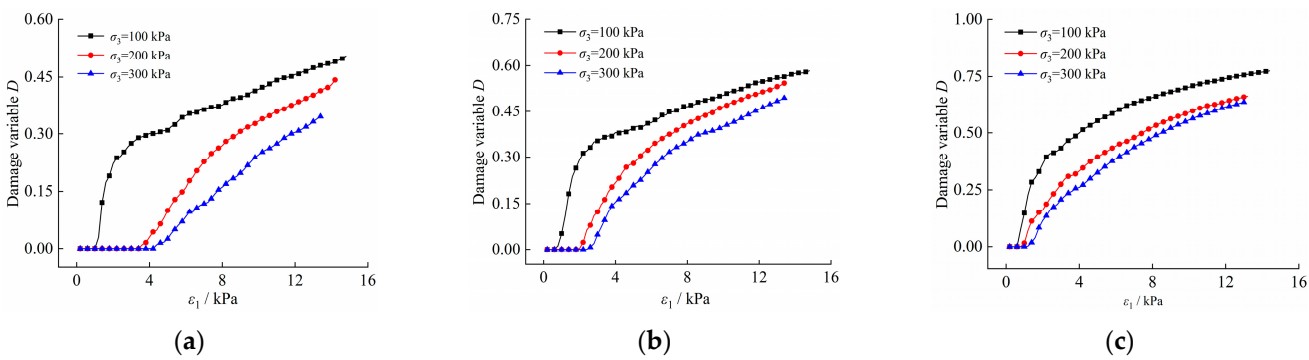

**Figure 7.** Variation pattern of the damage variable in the moistening and stress fields. (**a**) *s* = 50 kPa; (**b**) s = 100 kPa; (**c**) *s* = 200 kPa.

According to Figure 7: (i) As the strain increased, the damage variable *D* first increased rapidly at a larger rate. Although the rate of increase gradually declined, the overall increasing trend remained. However, *D* was below 1 throughout the test, i.e., the unsaturated soil exhibited incomplete damage. This occurred because, on the one hand, unsaturated soil is

prone to strain hardening, and its stress–strain under triaxial test conditions does not reach the limit state. On the other hand, unsaturated soil is frictional material, and the damaged part of the soil still has residual strength instead of completely losing its bearing capacity. Since this part of the bearing capacity is manifested in an undamaged form, the damage variable of unsaturated soil rarely reaches 1. (ii) In the moistening and stress fields, the damage variable increased with the increase in matrix suction. As the confining pressure increased, the damage variable decreased. This occurred because the soil always exerts cohesive strength first, followed by frictional strength in the unsaturated soil structure. Therefore, under the same strain conditions and constant total damage of unsaturated soil, the damage of the cohesive strength increases with the increase in matrix suction. However, the frictional strength increases with the increase in confining pressure, and the damage decreases accordingly. (iii) There is an initial damage threshold for unsaturated soil at the initial stage of damage, which decreases with the increase in matrix suction and increases with the increase in confining pressure. This occurs because, as the matrix suction increases, the viscoelasticity of the soil increases, and the hardening index $\alpha$ and the initial damage threshold of the soil decrease accordingly. The soil structural strength increases with the increase in confining pressure, and the initial damage threshold increases accordingly.

(2)  Laplace distribution parameters $m_0$ and $K_0$

The variation pattern of $m_0$ and $K_0$ under different matrix suction and confining pressure can be obtained from Table 2, as shown in Figures 8 and 9.

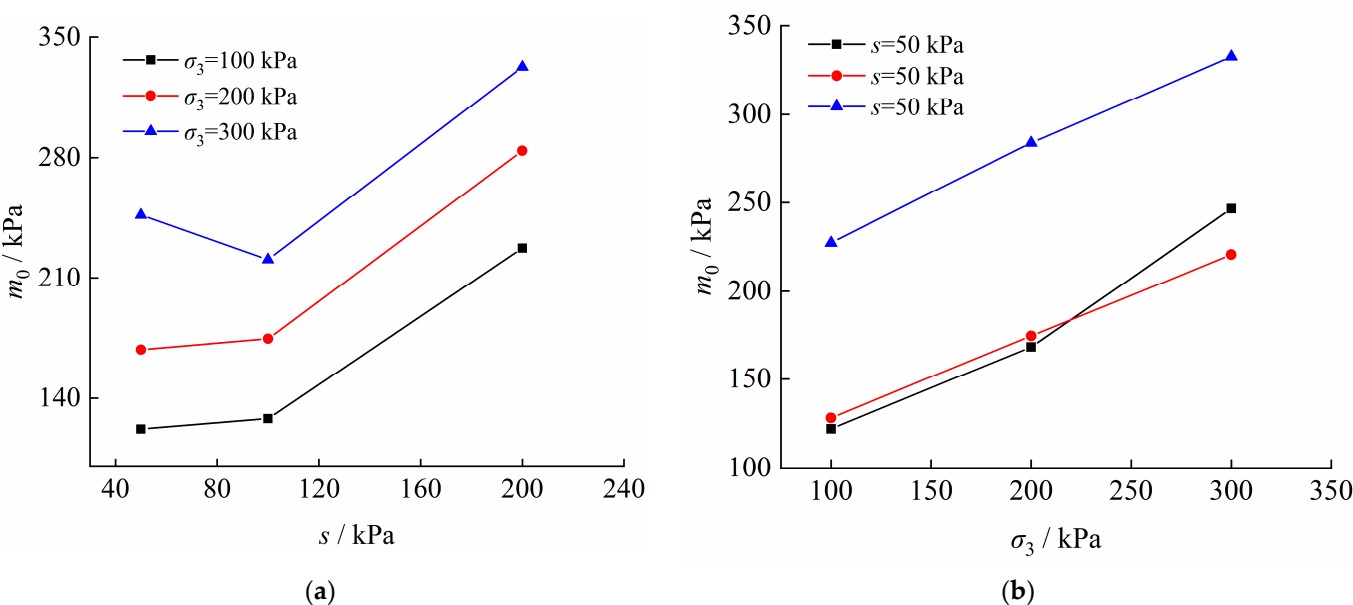

**Figure 8.** Variation pattern of $m_0$ in the moistening and stress fields. (**a**) $m_0 - s$; (**b**) $m_0 - \sigma_3$.

As shown in Figures 8a and 9a, $m_0$ and $K_0$ increase nonlinearly with the increase in matrix suction, indicating that the micro-unit strength decreases in the moistening field as the soil saturation increases. As shown in Figures 8b and 9b, $m_0$ and $K_0$ increase nonlinearly in the stress field with the increase of confining pressure, indicating that the micro-unit strength increases in the stress field as the confining pressure increases. Therefore, $m_0$ and $K_0$ performed well in reflecting the variation pattern of micro-unit strength in the moistening and stress fields.

(3)  Strain hardening parameter $\alpha$

The variation pattern of $\alpha$ under different matrix suctions and confining pressures is shown in Figure 10.

According to the definition of the hardening parameter $\alpha$ of unsaturated loess and Equation (32), the actual hardening capacity of unsaturated loess can be expressed as $(1 - \alpha)$.

As shown in Figure 10a, $(1 - \alpha)$ increases with the increase in matrix suction, indicating that the hardening capacity increases in the moistening field as the soil saturation increases. As shown in Figure 10b, $(1 - \alpha)$ increases with the increase in confining pressure, indicating that the hardening capacity increases in the stress field as the confining pressure increases. Therefore, hardening parameter $\alpha$ performed well in reflecting the hardening capacity of unsaturated loess in similar moistening and stress fields.

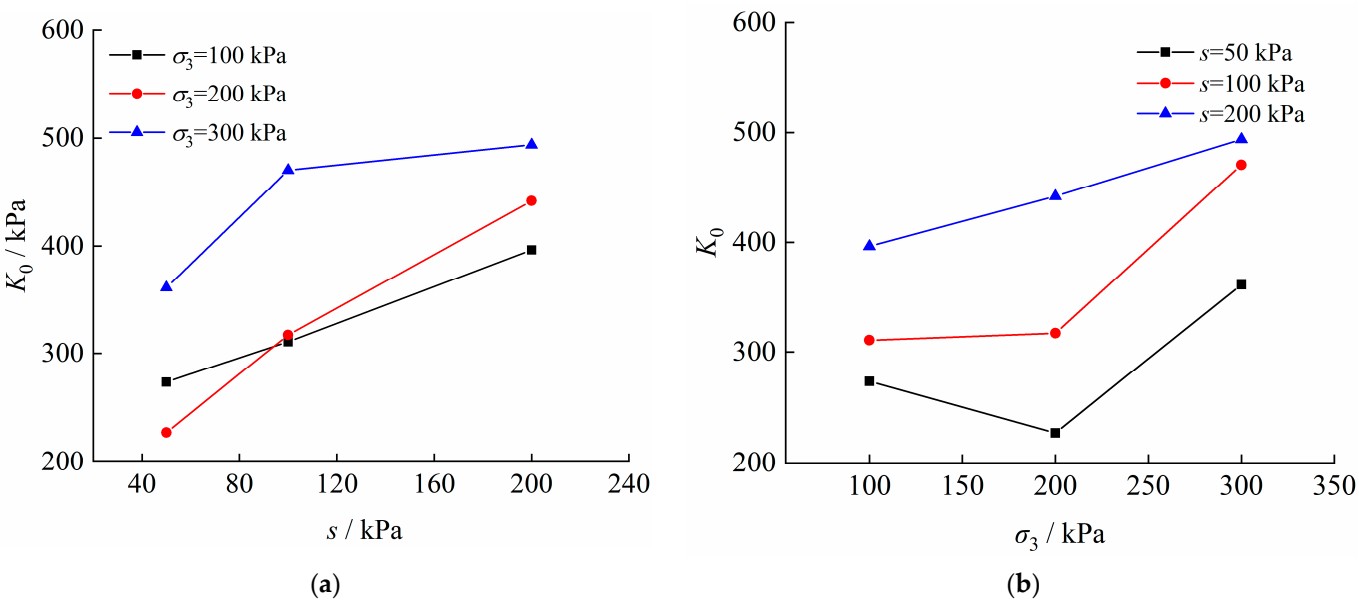

**Figure 9.** Variation pattern of $K_0$ in the moistening and stress fields. (**a**) $K_0 - s$; (**b**) $K_0 - \sigma_3$.

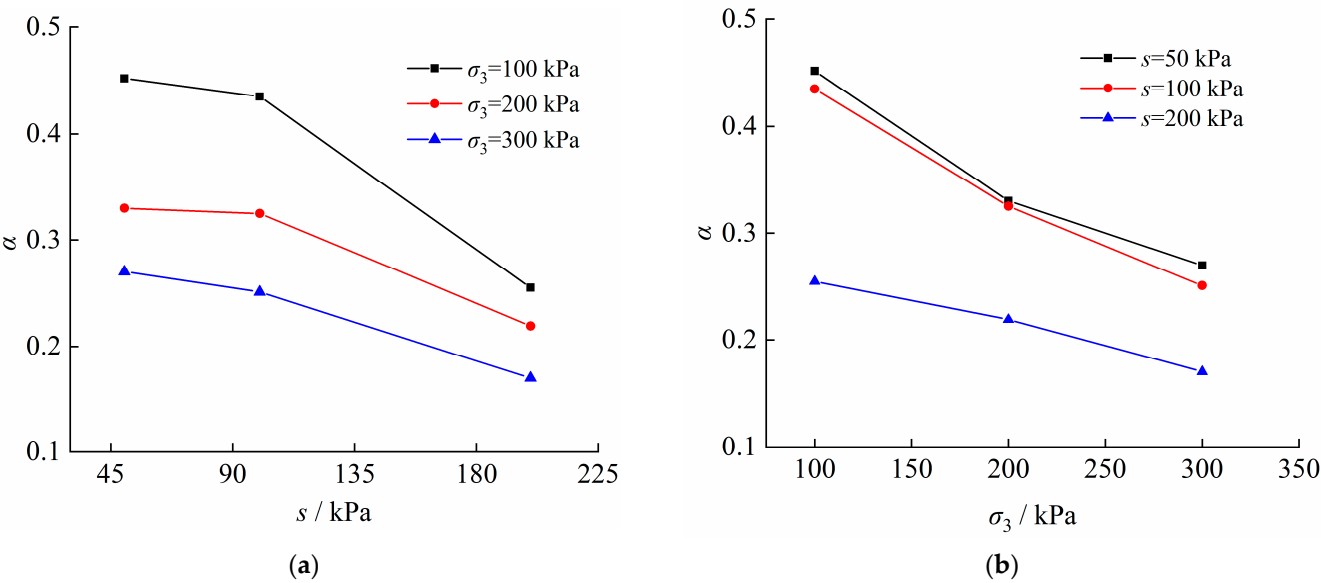

**Figure 10.** Variation pattern of $\alpha$ in moistening and stress fields. (**a**) $\alpha - s$; (**b**) $\alpha - \sigma_3$.

(4)    Elastic modulus $E_0$

The variation pattern of elastic modulus $E_0$ under different matrix suctions and confining pressures can be obtained from Table 2, as shown in Figure 11. According to the definition of soil viscoelasticity and the fractional calculus theory, soil viscoelasticity should be determined by both elastic modulus and hardening parameters.

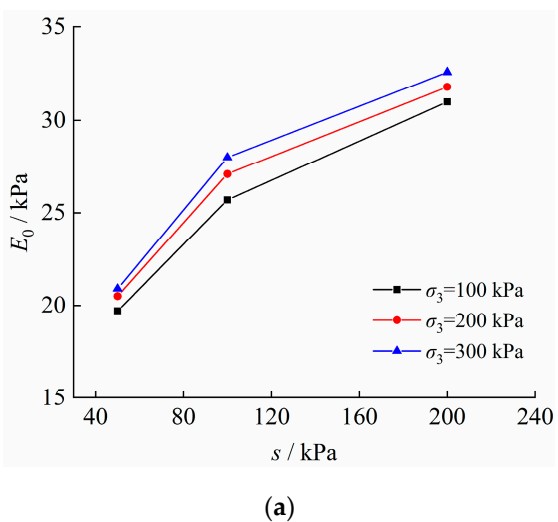

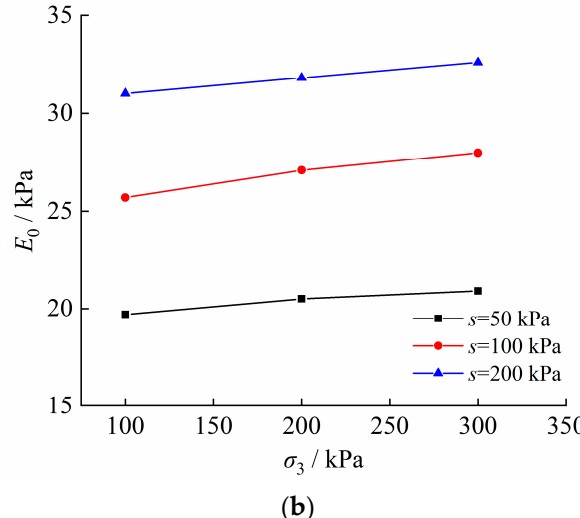

(a)

(b)

**Figure 11.** Variation pattern of $E_0$ in moistening and stress fields. (**a**) $E_0 - s$; (**b**) $E_0 - \sigma_3$.

As shown in Figure 11a, $E_0$ increases nonlinearly with the increase of matrix suction. Given the relationship between $\alpha$ and matrix suction shown in Figure 11a, the viscoelasticity decreases in the moistening field as the soil saturation increases. As shown in Figure 11b, $E_0$ increases linearly with the increase in matrix suction. Given the relationship between $\alpha$ and matrix suction shown in Figure 10b, the viscoelasticity increases in the stress field as the confining pressure increases. Therefore, elastic modulus $E_0$ and hardening parameter $\alpha$ performed well in reflecting the pattern of viscoelasticity of unsaturated loess in the moistening and stress fields.

## 5. Conclusions

Since water and load are two important factors affecting the mechanical properties of unsaturated soil, the damage variable evolution laws of unsaturated soil under the coupling effect of moistening and stress fields were explored. Based on the fractional calculus theory and the effective stress principle of unsaturated soil, a constitutive model of unsaturated soil was established. The proposed model can provide a scientific basis for the deformation and instability of unsaturated soil. The sensitivity of the main parameters of the proposed model was discussed, and the following conclusions were reached.

(1) Based on the classical soil mechanics principle, unsaturated soil mechanics and VG model, the mathematical formula for effective stress of unsaturated soil expressed by saturation is established in this paper. On this basis, the mathematical formula of mechanical shear strength of unsaturated soil, considering the coupling of moistening and stress fields, is established by Schrefler's shear strength formula.

(2) A fractional-order strain-hardening model based on a conventional unsaturated tri-axial shear test is established by using fractional-order calculus theory. Based on the principle of damage mechanics, the coupling damage variable of unsaturated soil under the influence of stress and water content is established. Finally, a fractional-order strain-hardening model of unsaturated soil under the coupling of stress and moistening fields is established.

(3) The experimental data of unsaturated loess with strain-hardening characteristics are used to verify the proposed model. Results show that the model can describe the whole process of strain-hardening of unsaturated soil under the coupling of stress and moistening fields.

(4) Through discussions on parameters of the proposed model, it is found that there is an initial threshold as unsaturated soil being damaged in coupling of the stress and moistening fields. The initial threshold value increases with the increase in confining pressure and decreases with the increase in matric suction. The hardening ability of

unsaturated soil increases with the increase in matric suction or confining pressure. The overall mechanical properties of unsaturated soils improve nonlinearly with the increase of matric suction or confining pressure.

The constitutive model proposed in this study was a discussion of the mechanical properties of unsaturated soil under the coupling effect of the moistening and stress fields, which could theoretically describe the stress–strain process. However, due to the lack of complete physical experimental data, it was difficult to discuss parameters $\delta$ and $\beta$ in the validation process. Further validation of the proposed model is required with experimental results of multiple groups of unsaturated soil.

**Author Contributions:** Conceptualization, H.Z.; Methodology, H.Z.; Resources, H.Z. and P.W.; Writing—original draft, P.W.; Writing—review & editing, H.Z.; Supervision, P.W.; Funding acquisition, H.Z. and P.W. All authors have read and agreed to the published version of the manuscript.

**Funding:** This research was financially supported by Sichuan Huaxi Group Co., Ltd. (No. HXKX2019/015, No. HXKX2019/019, No. HXKX2018/030), the Open Fund of Sichuan Provincial Engineering Research Center of City Solid Waste Energy and Building Materials Conversion and Utilization Technology (No. GF2022ZC009), and the Open Fund of Sichuan Engineering Research Center for Mechanical Properties and Engineering Technology of Unsaturated Soils (No. SC-FBHT2022-04).

**Institutional Review Board Statement:** Not applicable.

**Informed Consent Statement:** Not applicable.

**Data Availability Statement:** Some or all data, models, or code that support the findings of this study are available from the corresponding author upon reasonable request. All data shown in the figures and tables can be provided on request.

**Conflicts of Interest:** The authors declare no conflict of interest.

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
