# Peer review of "Fractional Calculus-Based Statistical Damage Model of Unsaturated Soil under the Coupling Effect of Moistening and Stress Fields"

_applsci, doi:10.3390/app13169156_

Round 1
Reviewer 1 Report
The paper is interesting and within the scope of the journal. However, there are some comments need to be addressed before it gets be published. They are provided as follows:
1. The literature review needs to be cover all previous and recent studies on the scope of the paper.
2. The necessities and novelties of this study should be highlighted in the last paragraph of Introduction section of this paper.
3. The validation of the model is performed based on stress amplitude of 100, 200 and 300 kPa. So, the question is: what is accuracy of the model when the stress amplitude is out of range of 100 to 300 kPa? For instance, if the stress amplitude is 600 kPa, how you can prove the model validation? It should be comprehensively elaborated and discussed in the revised paper.
4. More elaborations on the results should be provided in the revised paper.
There is no comment on this issue.
Author Response
Dear Editors and Reviewer:
Thank you for your letter and for the reviewer’ comments concerning our manuscript entitled “Fractional Calculus-based Statistical Damage Model of Unsaturated Soil under the Coupling Effect of Moistening and Stress Fields” (ID: applsci-2429129). Those comments are all valuable and very helpful for revising and improving our paper, as well as the important guiding significance to our researches. We have studied comments carefully and have made correction which we hope meet with approval. Revised portion are marked in red in the paper. The main corrections in the paper and the responds to the reviewer’s comments are as flowing:
Responds to the reviewer’s comments:
Reviewer #1:
1 Response to comment: The literature review needs to be cover all previous and recent studies on the scope of the paper.
Response: Considering the Reviewer’s suggestion, the explanation is as follows:
Based on the basic theories of soil mechanics, fractional-order calculus, statistical damage principle and unsaturated soil mechanics, this paper provides a new model to describe the mechanical mechanism of strain hardening of unsaturated soil under the influence of water and stress. For each basic theory above mentioned, there are too many articles relevant to it to cite all of them. In this paper, some representative literatures have been selected for references.
2 Response to comment: The necessities and novelties of this study should be highlighted in the last paragraph of Introduction section of this paper.
Response: We have re-written this part in the last paragraph of Introduction section according to the Reviewer’s suggestion.
3 Response to comment: The validation of the model is performed based on stress amplitude of 100, 200 and 300 kPa. So, the question is: what is accuracy of the model when the stress amplitude is out of range of 100 to 300 kPa? For instance, if the stress amplitude is 600 kPa, how you can prove the model validation? It should be comprehensively elaborated and discussed in the revised paper.
Response: Considering the Reviewer’s suggestion, the explanation is as follows:
The mechanical properties of unsaturated soil under 100~300kPa confining pressure are verified in this paper. This paper verifies the confining pressure most likely involved in the project in the unsaturated area, which is also an example verification. It is also not possible to verify all confining pressures in a single paper. For the mechanical properties of unsaturated soil under other confining pressures, more test data are needed for further verification. In response to the reviewer's comments, the author of the manuscript provides explanations in page 11, lines 306~308.
4 Response to comment: More elaborations on the results should be provided in the revised paper.
Response: We have re-written this part according to the Reviewer’s suggestion in page 14, lines 388~409.
Special thanks to you for your good comments.
We tried our best to improve the manuscript and made some changes in the manuscript. These changes will not influence the content and framework of the paper. And here we did not list the changes but marked in red in revised paper.
We appreciate for Editors/Reviewer’ warm work earnestly, and hope that the correction will meet with approval.
Once again, thank you very much for your comments and suggestions.

Reviewer 2 Report
Check comments in the attached file.

Although the use of English is fairly good, a professional editing is strongly recommended.
Author Response
Dear Editors and Reviewer:
Thank you for your letter and for the reviewer’ comments concerning our manuscript entitled “Fractional Calculus-based Statistical Damage Model of Unsaturated Soil under the Coupling Effect of Moistening and Stress Fields” (ID: applsci-2429129). Those comments are all valuable and very helpful for revising and improving our paper, as well as the important guiding significance to our researches. We have studied comments carefully and have made correction which we hope meet with approval. Revised portion are marked in red in the paper. The main corrections in the paper and the responds to the reviewer’s comments are as flowing:
Responds to the reviewer’s comments:
Reviewer #2:
- Response to comment: English need to review
Response: Considering the Reviewer’s suggestion, We have checked and revised the English text of the manuscript.
- Response to comment: Explain “ By the time of writing”
Response: This expression expresses a period of time. More specifically: up to now.
- Response to comment: What was missing was talking about the genesis of soils and the relationship with non-saturation and its effects on stress-strain.
Response: Considering the Reviewer’s suggestion, we have added an explanation of unsaturated soils in page 2 of the manuscript, lines 70-73.
- Response to comment: what means: sound methods
Response: We are very sorry for our incorrect writing “sound methods”. The correct expression is “reasonable method”.
- Response to comment: What are the engineering issues associated with unsaturated soils?
Response: Unsaturated soil engineering includes: slope engineering, foundation pit engineering and other related projects, the main problems generally include unsaturated soil rainfall seepage, deformation, strength, stability and other problems.
- Response to comment: In situations of unsaturated soils, imagining that there will be a high increase in moisture content, what relevance do the authors consider in this case?
Response: Considering the Reviewer’s suggestion, the explanation is as follows:
The model in this paper analyzes the moisture content in the common range in general engineering. For the special condition of high moisture content, soil-water characteristic test should be carried out under the condition of high matric suction to clarify the rule of matric suction under high moisture content. The test results are applied to the model in this paper according to equations (7), (10) and (38), which can realize the application of the model in the case of high moisture content.
- Response to comment: Why 15%? in Fig 2.
Response: In the "Standard for Geotechnical Test Methods" (GBT50123--2019)(in chinese), it is stipulated that for strain-hardened soil, the stress when the strain is 15% is taken as the shear strength.
- Response to comment: Some simbols definitions from the equations are missing. the authors must define all of then or a list or an end of paragraph
Response: represents the gamma function, which is defined in equation (12).
- Response to comment: What means each symbol in Table 1.
Response: We have re-written this part according to the Reviewer’s suggestion.
- Response to comment: For better understanding to help analyzes put these three graphs together.
Response: We have made correction according to the Reviewer’s comments.
- Response to comment: Explain how the modulus was determined. Graphically or using mathematical method?
Response: The modulus E0 is obtained from the conventional triaxial test curve.
- Response to comment: If the authors have test data in the saturated condition, it would be interesting to include it in the article, which will greatly improve the analyses.
Response: Considering the Reviewer’s suggestion, we are very sorry that we cannot give the test data in the saturated state in a short time. But, we will prepare and carry out this work in accordance with the comments of the reviewers.
- Response to comment: What about the effect of soil stiffening in view of high suctions? What can change the modulus of deformability?
Response: According to the rule analysis of E0 in this paper, with the increase of matric suction, E0 will increase, but the increase rate will gradually decrease. Therefore, when the matric suction increases to a certain extent, E0 will remain stable. Due to the limited experimental data in this paper, this stable value cannot be found, which will be analyzed in subsequent studies.
- Response to comment: There is a lot of data that has not been analyzed, which does not allow for a more in-depth conclusion. The authors are requested to rewrite the conclusions, going deeper into the data and the proposal.
Response: We have re-written this part according to the Reviewer’s suggestion in page 14 of the manuscript, lines 388-409.
Special thanks to you for your good comments.
We tried our best to improve the manuscript and made some changes in the manuscript. These changes will not influence the content and framework of the paper. And here we did not list the changes but marked in red in revised paper.
We appreciate for Editors/Reviewer’ warm work earnestly, and hope that the correction will meet with approval.
Once again, thank you very much for your comments and suggestions.
